# Treating Onychomycosis with Efinaconazole: Considerations for Diverse Patient Groups

**DOI:** 10.3390/jof11120843

**Published:** 2025-11-28

**Authors:** Aditya K. Gupta, Daniel Taylor, Daniel Dennis, Tong Wang, Elizabeth A. Cooper

**Affiliations:** 1Division of Dermatology, Department of Medicine, Temerty Faculty of Medicine, University of Toronto, Toronto, ON M5S 1A1, Canada; 2Mediprobe Research Inc., London, ON N5X 2P1, Canada; dtaylor@mediproberesearch.com (D.T.); ddennis@mediproberesearch.com (D.D.); twang@mediproberesearch.com (T.W.); lcooper@mediproberesearch.com (E.A.C.)

**Keywords:** onychomycosis, tinea unguium, nail fungus, efinaconazole, comorbidity

## Abstract

Onychomycosis is a common nail disease that manifests with varying severity and frequency in specific patient populations, warranting a personalized treatment approach. Novel topical antifungals, such as efinaconazole 10% approved for use in North America and Japan, offer a safe treatment option for many of these patients, though real-world use requires special considerations. In this scoping review, a literature search was conducted in October 2025 using PubMed, Embase (Ovid), the Cochrane Central Register of Controlled Trials (CENTRAL), and Web of Science (Core Collection). In children and adolescents (≥6 years), efinaconazole 10% has shown higher efficacy rates than in adults, possibly attributed to less nail trauma, thinner nail plates, and faster nail growth. In the elderly, a mycological response can precede visual nail improvements, which may require extending treatment beyond the standard 48-week regimen, along with intermittent maintenance therapies. Although antifungal resistance is a concern, dermatophytes—including terbinafine-resistant strains—have generally shown high susceptibility to efinaconazole. In diabetic individuals, onychomycosis should be treated promptly to prevent secondary complications. Efinaconazole 10% showed similar efficacy in this population, regardless of glycemic control. In historically underserved populations, efinaconazole 10% showed no significant difference in efficacy for Latino/Hispanic patients, though further research is needed. Overall, efinaconazole 10% solution was well-tolerated across patient groups, with application-site reactions occurring without systemic sequalae. Healthcare providers are advised to check for concomitant tinea pedis, which increases the risk of relapse or re-infection, and advise patients on nail polish use, which may degrade after topical antifungal application. A shared decision-making framework can help improve treatment compliance and patient satisfaction.

## 1. Introduction

Onychomycosis, a common fungal infection of the nails, affects up to 10% of the global population and is particularly prevalent among older adults, individuals with diabetes, and immunocompromised patients [1,2]. Characterized by nail discoloration, thickening, and onycholysis, onychomycosis often leads to discomfort, functional impairment, and psychosocial distress. While oral antifungals such as terbinafine and itraconazole are effective, their use is limited by systemic side effects, hepatic dysfunction, and drug–drug interactions [3,4,5]. Toenail onychomycosis, in particular, is often regarded as more difficult to treat due to higher likelihoods of trauma, secondary infections, and slower nail growth. These concerns are especially relevant in medically complex patients.

Topical antifungals are commonly used for managing mild-to-moderate onychomycosis, as well as for whom oral antifungals are contraindicated. In particular, efinaconazole 10% solution—a triazole medication approved in 2014 by the U.S. Food and Drug Administration (FDA)—is being increasingly prescribed and accounts for approximately 15% of Medicaid reimbursements for treatments of superficial fungal infection [6,7]. This agent is also approved for use in Canada and Japan [8,9], and is currently being investigated for pediatric use by the European Medicines Agency (EMA) [10]. Despite its demonstrated efficacy and safety, patient response to efinaconazole can be influenced by differences in nail physiology, immune function, comorbidities, and fungal burden.

In this review, we discuss considerations for onychomycosis management in children, the elderly, diabetic individuals, racial/ethnic minorities, and patients with concomitant tinea pedis, focusing on the current evidence for efinaconazole as the first-line topical therapy. We also explore concerns regarding nail polish use affecting drug penetration, the utility of long-term and maintenance treatments, as well as the in vitro activity of efinaconazole against dermatophytic fungi, including terbinafine-resistance strains.

## 2. Materials and Methods

A scoping review was conducted in reference to the PRISMA guideline, aiming to compile and present evidence on the use of efinaconazole 10% for onychomycosis in diverse patient populations, as well as its in vitro activity against dermatophytes. To identify clinical studies, an electronic literature search was conducted on 6 October 2025 across PubMed, Embase (Ovid), and the Cochrane Central Register of Controlled Trials (CENTRAL). The search terms included ‘efinaconazole’, ‘JUBLIA’, ‘CLENAFINE’, ‘KP-103’, and ‘IDP-108’, which were combined with ‘onychomycosis’, ‘tinea unguium’, or ‘nail fungus’. To identify in vitro susceptibility studies, a second electronic literature search was conducted on 7 October 2025 across PubMed, Embase (Ovid), and Web of Science (Core Collection). The search terms included ‘efinaconazole’, ‘JUBLIA’, ‘CLENAFINE’, ‘KP-103’, and ‘IDP-108’, which were combined with ‘minimum inhibitory concentration’, ‘antifungal susceptibility testing’, ‘in vitro susceptibility’, ‘CLSI’, or ‘EUCAST’. No date restriction was applied. The inclusion criteria were as follows: (1) clinical studies that treated onychomycosis patients with a mycology-confirmed diagnosis, using efinaconazole in its U.S. FDA-approved formulation (10% solution), or (2) studies that tested the in vitro dermatophyte susceptibility to efinaconazole using the standard broth microdilution method. Conference proceedings, reviews, and expert opinions were excluded.

## 3. Results and Discussion

Nine studies, involving a total of 1307 patients, examined the use of efinaconazole 10% in specific patient populations diagnosed with onychomycosis (Table 1) [11,12,13,14,15,16,17,18,19]. Of these, four studies described post hoc analyses of two pivotal, phase 3 trials published by Elewski et al. [12,15,17,19,20]. The following sections discuss available data on children (Section 3.1), the elderly (Section 3.2), diabetic patients (Section 3.3), and Hispanic/Latino patients (Section 3.4). Two treatment endpoints were used for comparison: (1) mycological cure, defined as negative direct microscopic examination and negative fungal culture, and (2) complete cure, defined as complete clinical resolution in addition to mycological cure. The in vitro susceptibility of dermatophytes to efinaconazole is examined and compared against other standard antifungals in Section 3.5. The utility of an intermittent maintenance regimen, as well as concerns for concomitant tinea pedis and nail polish use, are discussed in Section 3.6 and Section 3.7.

### 3.1. Children

The risk of hepatotoxicity with systemic antifungals and need for laboratory monitoring present a concern for the pediatric use of systemic antifungals; additionally, the administration of oral tablets/capsules to young children can be difficult [21]. Topical therapy can provide a more favorable route of treatment for pediatric onychomycosis, while also minimizing treatment risk. Children also generally exhibit faster nail growth and thinner nails compared to adults, which may enhance topical drug penetration and shorten the time to clinical improvement [22].

Efinaconazole 10% topical solution is one of the few topical antifungals that have been U.S. FDA-approved for use in children 6 years old and up [6]. In a phase 4 trial, 62 children or adolescents (6–16 years) were enrolled to receive efinaconazole 10% once daily for 48 weeks [11]. Efficacy results, based on 60 patients who received at least one dose of the study drug, showed a continuous increase in complete and mycological cure rates through week 52 [11]. Specifically, complete cure rates were 0% (0/60) at week 12 and 40% (24/60) at week 52. Mycological cure rates were 36.7% (22/60) at week 12 and 65% (39/60) at week 52. These cure rates were significantly higher compared to the phase 3 adult population (median age: 52–54 years) receiving the same regimen of efinaconazole 10%, where complete cure rates were 15–18% and mycological cure rates were 53–55% [20].

Some of these differences in efficacy rates between children and adults can be attributed to higher likelihoods of pre-existing nail trauma in older adults, which impedes nails returning to a normal appearance, despite a mycological response. Treatment was well-tolerated, with two patients experiencing ingrown nails related to treatment [20]. No study discontinuations occurred due to treatment-emergent adverse events. The low systemic exposure risk was demonstrated in a pharmacokinetic sub-study, where peak serum concentrations (C_max_) were 0.5 ± 0.3 ng/mL for efinaconazole and 1.6 ± 1.3 ng/mL for the H3 metabolite by day 28 [20].

### 3.2. Elderly

Onychomycosis shows an age-dependent increase in incidence, with elderly individuals having approximately a fivefold higher risk than the general population [2]. This translates to unique treatment challenges in routine practice, including slower nail outgrowth, comorbidities, and polypharmacy. As a result, topicals such as efinaconazole 10% have been extensively studied in this population due to their favorable safety profile. However, patients may experience difficulties with the regular self-application of topical medications. Ideally, these factors should be addressed under a shared decision-making framework to increase compliance when topical medications are prescribed.

In a post hoc analysis of two phase 3 trials, 218 patients aged 65–71 years demonstrated a significantly higher complete cure rate of 13.6% (22/162) with efinaconazole 10% compared to the vehicle (3.6% [2/56]). Additionally, efinaconazole 10% showed a significantly higher mycological cure rate of 59.3% (96/162) compared to the vehicle (12.5% [7/56]) [12,13]. Efinaconazole 10% or the vehicle were applied once-daily for 48 weeks. No significant difference in adverse event incidence was observed between the efinaconazole and vehicle arms, with application site dermatitis occurring in 4.3% (7/162) of patients receiving efinaconazole 10%.

For elderly patients who may experience more extensive onychomycosis, Iozumi et al. evaluated efinaconazole 10% applied once daily for up to 72 weeks, or until complete cure, in 118 elderly individuals, including those with >50% nail area involvement [14]. The results showed that more than half of the patients achieved mycological cure (60.2% [71/118]), while 33.1% (39/118) achieved complete cure. However, these findings are not directly comparable to other investigations due to differences in defining mycological cure (i.e., negative KOH microscopic examination alone without fungal culture). Similarly to other studies, application-site reactions (e.g., contact dermatitis, erythema) occurred in patients receiving efinaconazole, without systemic complications [14].

In practice, this patient population may often require an extension of the standard 48-week regimen. A phase 4 trial evaluated the use of efinaconazole 10% for up to 2 years, where the majority of patients were 60 years or older [23]. Of the 69 patients who completed treatment, an increase in mycological response and effective cure (≤10% clinical improvement and mycological cure) was observed from year 1 to year 2, across both younger and older age groups. However, increased nail thickness was identified as a negative prognostic factor, which may necessitate routine trimming or filing in conjunction with topical efinaconazole 10% applications [23].

### 3.3. Diabetic Patients

Onychomycosis prognosis in diabetic patients is often worsened by reduced vascular supply, hyperglycemia-induced immunosuppression, delayed nail growth, and poor foot care [24,25,26]. These factors may slow treatment response, increase recurrence risk, and elevate the risk of complications such as foot ulcers and cellulitis. Similarly to the elderly population, comorbidities and polypharmacy associated with diabetes complicate the risks of oral antifungal use, making effective topical antifungals appealing for use in this group.

It has been hypothesized that high glucose levels increase the risk of onychomycosis and treatment resistance. In an open-label study, Shofler et al. assessed the efficacy of efinaconazole 10% in 40 onychomycosis patients with well-controlled or uncontrolled diabetes (mean HbA1C: 7.7 ± 1.6) [16]. Among these patients, 47.5% (19/40) were receiving insulin injections, 67.5% (27/40) were receiving oral hypoglycemics, and 10% (4/40) were undergoing dialysis. After applying efinaconazole 10% once-daily for 50 weeks, the complete cure rate—based on treatment completion—was 11.1% (4/36), and the mycological cure rate was 58.3% (21/36). Importantly, no correlation was found between HbA1c levels and efficacy results [16]. Application-site reaction (i.e., vesicles of the target toenail) occurred in 2 patients without systemic sequalae. Additionally, one case of leg cellulitis and one case of verruca were reported.

These findings corroborated a post hoc analysis of 112 patients with well-controlled diabetes from two phase 3 trials, which reported a complete cure rate of 13% and a mycological cure rate of 56.5% [15]. Patients were treated with efinaconazole 10% once-daily for 48 weeks. Adverse reactions occurred at similar rates in both the efinaconazole and the vehicle groups, with two patients discontinuing due to application site erythema.

### 3.4. Racial and Ethnic Groups

Achieving equitable access to dermatology care is an ongoing effort. For onychomycosis, historically underserved populations face a higher risk of contracting the disease [27]. This increased risk may be linked to socioeconomic status, comorbidities such as obesity and diabetes, and rural communities lacking access to dermatology care [28,29]. With efinaconazole 10% treatment, available evidence demonstrates a similar efficacy profile across racial and ethnic groups, although further studies are needed.

In a post hoc analysis of two phase 3 trials, efficacy of efinaconazole 10% was compared between Latino patients (N = 270) and non-Latino patients (N = 1380) [17]. Across both groups, patients receiving efinaconazole 10% once daily for 48 weeks achieved significantly higher complete and mycological cure rates than the vehicle. Furthermore, the complete cure rate was higher in Latino patients than non-Latino patients (25.6% [69/270]) vs. 17.2% [237/1380]), as well as the mycological cure rate (61.5% [166/270]) vs. 55.4% [764/1380]). Of note, both patient groups had similar proportions of individuals with diabetes and similar body weight; however, the Latino patient group tended to be younger and had a higher proportion of females [17]. Adverse reactions were deemed mild to moderate, which resolved without sequelae.

These findings were corroborated in a phase 2 study of 135 Hispanic/Latino patients with onychomycosis [18]. Enrolled patients were also predominately female (54.1% [73/135]), and were randomized to receive three different efinaconazole formulations or the vehicle. After receiving once-daily applications for 36 weeks, patients receiving the approved efinaconazole 10% formulation achieved a complete cure rate of 25.6% (10/39) and a mycological cure rate of 87.2% (34/39) [18]. Four application-site reactions were deemed treatment-related including blister, contact dermatitis, erythema, and ingrown nail.

### 3.5. In Vitro Activity of Efinaconazole Against Dermatophytes

As an inhibitor of sterol 14α-demethylase, efinaconazole disrupts ergosterol biosynthesis—a building block of the fungal cell membrane—resulting in growth arrest, morphological changes, and eventual fungal cell death [30]. Inhibition of 14α-demethylase results in the accumulation of toxic sterol intermediates (4,4-dimethylsterols, 4α-methylsterols), which compromises cell membrane integrity [30]; this effect is demonstrated in dermatophyte hyphae, which exhibit dysmorphic changes (flattening, cell membrane separation, organelles degeneration) in a dose-dependent manner [30].

In recent years, a concerning trend of antifungal resistance has been reported worldwide, highlighting the need for novel antifungals. Clinically, isolates obtained from onychomycosis treated with efinaconazole 1% for 48 weeks showed no significant differences in in vitro susceptibility between pre- and post-treatment time points [31]. Prolonged in vitro exposure of *Trichophyton rubrum* and *T. mentagrophytes* to efinaconazole similarly did not result in significant changes in minimum inhibitory concentrations (MICs) [32]. However, resistance development has been demonstrated under certain laboratory conditions, possibly due to the overexpression of *ERG11A*, which encodes an isozyme of 14α-demethylase [33]. Azole resistance can also develop through efflux pumps [34].

Eighteen studies were included that examined the susceptibility profile of dermatophytes to efinaconazole, as well as other relevant antifungals in the treatment of onychomycosis (amorolfine, ciclopirox, fluconazole, griseofulvin, itraconazole, ketoconazole, luliconazole, ravuconazole, tavaborole, and terbinafine), all of which were conducted using the same set of isolates [31,35,36,37,38,39,40,41,42,43,44,45,46,47,48,49,50,51]. The most commonly reported growth inhibition endpoint was MIC_90_ (≥90% inhibition), determined according to the Clinical & Laboratory Standards Institute (CLSI) protocol. Based on this endpoint, a total of 2912 dermatophyte isolates were tested in vitro against efinaconazole. Of these, 61.8% (1799/2912) were *T. rubrum*, 20.5% (597/2912) were the *T. mentagrophytes* complex including *T. mentagrophytes*, *T. interdigitale*, and *T. indotineae*, 9.1% (264/2913) were other *Trichophyton* species, 7.1% were *Microsporum canis* (208/2912), 1.5% were *Nannizzia gypsea* (43/2912), and 0.03% were *Epidermophyton floccosum* (1/2912).

The proportions of dermatophyte isolates corresponding to each MIC_90_ category are presented in Table 2. Overall, efinaconazole demonstrated a favorable efficacy profile with 78.4% (2284/2912) of isolates demonstrating MIC_90_ of ≤0.03 µg/mL, a finding similar to that of luliconazole (99.9% [973/974]). In comparison, the majority of isolates tested against ravuconazole exhibited an MIC_90_ of 0.06 µg/mL (49.6% [131/264]), while all isolates tested against ketoconazole exhibited an MIC_90_ of 0.125 µg/mL. Amorolfine exhibited a bimodal susceptibility pattern, with 38.8% (259/668) and 41.3% (276/668) of isolates showing an MIC_90_ of ≤0.03 µg/mL and 2 µg/mL, respectively. This finding reflects the existence of both susceptible and non-susceptible subpopulations. For itraconazole, a varied dermatophyte susceptible profile was observed, with MIC_90_ ranging from 0.06 to 1 µg/mL. By contrast, a predominate non-susceptible population was apparent when examining ciclopirox, fluconazole, griseofulvin, and tavaborole. A strong propensity for dermatophyte resistance against fluconazole was evident, as the majority of isolates showed an MIC_90_ of 16 µg/mL.

In recent years, the management of dermatophytosis has been increasingly challenged by the emergence of terbinafine resistance [52], which is one of the most consumed systemic antifungals globally alongside itraconazole and fluconazole [53]. Topical terbinafine 1% is also classified as an essential medicine by the World Health Organization for the treatment of skin fungal infections [54]. In the context of onychomycosis, a personalized approach to treatment selection has been emphasized [55], particularly given that long-term treatments (≥6 months) are often required for healthy toenails to fully grow out. Such prolonged treatments may inadvertently create conditions for resistance development, as seen with terbinafine. Two subgroups were analyzed based on the terbinafine susceptibility profile (MIC_90_; ≤0.03 µg/mL vs. ≥0.25 µg/mL) (Table 2). In studies where molecular investigations were conducted, a low in vitro susceptibility to terbinafine was attributed to single nucleotide variations of the gene encoding its drug target—squalene epoxidase (SQLE)—as part of the fungal cell membrane synthesis pathway [36,37,47,48]. Specifically, these *SQLE* mutations were located at the following amino acid loci: 393 (L393F, L393del), 394 (Y394del), 397 (F397L).

Among terbinafine susceptible dermatophyte isolates, amorolfine, efinaconazole, and luliconazole demonstrated a similar degree of in vitro efficacy with 93–100% of isolates showing an MIC_90_ of ≤0.03 µg/mL, while fluconazole resistance was still evident (Table 2). Among terbinafine non-susceptible isolates, most still demonstrated high susceptibility (MIC_90_: ≤0.03 µg/mL) to luliconazole (99.7% [304/305]), followed by efinaconazole (50.6% [199/393]). In the absence of clinical breakpoints, a cut-off value of 0.25 µg/mL was applied for comparative analysis between antifungals (Figure 1). Using this approach, efinaconazole, ketoconazole, luliconazole and ravuconazole demonstrated a similar efficacy profile in terbinafine non-susceptible dermatophyte isolates. When contrasting terbinafine susceptible and non-susceptible dermatophyte isolates, evidence of cross-resistance was observed for amorolfine, ciclopirox, and itraconazole (Table 2). The MIC_90_ for amorolfine was notably higher in terbinafine non-susceptible isolates than susceptible isolates (2 µg/mL vs. ≤0.03 µg/mL). A similar trend was observed for ciclopirox where the MIC_90_ ranged from 0.25 to 0.5 µg/mL in terbinafine susceptible isolates, and from 1 to 2 µg/mL in non-susceptible isolates. For itraconazole, the majority of terbinafine non-susceptible isolates (40.1% [124/309]) had an MIC_90_ of 1 µg/mL, whereas the majority of terbinafine susceptible isolates (48.2% [201/417]) had an MIC_90_ of 0.125 µg/mL.

### 3.6. Long-Term and Prophylactic Treatment

While resistance may theoretically arise during prolonged treatment, especially concerning sub-optimal dosing at levels below the MIC, the current available data from dermatophytes do not indicate a significant risk of resistance development against efinaconazole. A potent topical agent should be considered for treating vulnerable populations, whose management is often complicated by comorbidities, polypharmacy, and slow nail outgrowth. In onychomycosis, the patient population in clinical trials often does not reflect real-world scenarios, where treatment relapse or re-infection can occur at a rate of 25% [56]. Due to the difficult-to-treat nature of onychomycosis, it is not uncommon for efinaconazole treatments to extend beyond the standard 48-week schedule in routine practice.

In a real-world study of onychomycosis patients with varying ages (<65 years and ≥65 years) and disease severities (ranging from 20% to ≥50% target nail area affected), efinaconazole 10% was applied once daily until complete cure or for up to 72 weeks [14]. Of the 219 patients enrolled, complete cure was achieved in 31.1% (68/219) of patients, with an upward trend over time, from week 48 to week 60 then week 72, regardless of disease severity [14]. A similar trend was seen in mycological cure rates, which peaked at week 36 and were maintained through the end of study (61.6% [135/219]) [14]. Treatment-related adverse events occurred at a rate of 6.4% (13/219) attributed to applicate site reactions, with the most common being contact dermatitis (5% [11/219]) [14]. No serious adverse events were deemed treatment-related.

After completing efinaconazole treatment, the utility of an intermittent maintenance regimen was evaluated in a single-center Canadian study [57]. A total of 35 patients, who completed an initial treatment phase of efinaconazole 10% applied once daily for 72–96 weeks, were subject to an additional period of intermittent efinaconazole treatment for 96 weeks [57]. Patients were instructed to apply efinaconazole 10% for 2 or 3 days per week at their discretion. At the end of the study, all 6 patients who achieved clinical improvements during the treatment phase had a sustained mycological response during the maintenance phase, with one patient achieving mycological cure during the maintenance phase. For patients with >10% affected nail area remaining after the treatment phase, 10.3% (3/29) showed significant clinical improvement. One patient developed mild-to-moderate application site reactions during the maintenance phase. No treatment-related adverse reactions led to treatment discontinuation.

### 3.7. Other Treatment Considerations

The use of any topical therapy for onychomycosis can be challenging, as it requires significant patient commitment to frequent, long-term application periods. Despite good mycological cure rates, the rates of clinical cure lag behind oral therapies and clinical improvements may be particularly challenging for vulnerable populations where comorbidities may delay nail outgrowth rate and/or increase nail dystrophy symptoms. We present here some treatment considerations that may be useful for improving efinaconazole compliance and efficacy.

For patients with immunodeficiency or diabetes, tinea pedis concomitant with onychomycosis can be a prevalent problem, with each condition being a reservoir for recurrent or ongoing infection. When both infections are present, the treatment of tinea pedis may improve efinaconazole treatment outcomes for onychomycosis.

In a phase 3 trial of onychomycosis patients receiving efinaconazole 10% treatment, interdigital tinea pedis was diagnosed in 21.3% (352/1655) of patients, which investigators could optionally treat by topical antifungals (butenafine, luliconazole, ketoconazole) [19]. Among patients whose tinea pedis was also treated, the complete cure rate for onychomycosis was 29.4% (40/136), compared to 16.1% (15/93) in those whose tinea pedis was left untreated [19]. Although this post hoc analysis is limited by the lack of data on tinea pedis treatment compliance, it is conceivable that the timely treatment of tinea pedis could prevent its dissemination to adjacent nails, thus improving the treatment response for onychomycosis.

Onychomycosis patients may apply nail polish to mask dystrophic nail changes, which can potentially interfere with the penetration of topical antifungals. In these instances, patients should be advised to avoid using polish, or they may choose to reduce their compliance, which could adversely impact clinical outcomes. For efinaconazole, available data does not indicate that nail polish reduces drug penetration; however, the appearance and texture of the polish may be altered. In ex vivo studies, efinaconazole was applied to polished fingernails for 7 days, resulting in discoloration and texture changes, as well as color-transfer to the brush applicator and solution [58,59]. Despite these cosmetic changes, the degree of drug permeation through the nail was not significantly different from unpolished nails [59]. These findings were corroborated in a clinical study where female patients, with or without nail polish, were treated with efinaconazole 10% once-daily for 48 weeks [60]. At week 52, no significant difference in clinical improvement was observed, although patients reported dissatisfaction due to changes in polish color and texture. To mitigate these cosmetic issues, patients can be advised to apply a darker-colored polish and apply both a top and base coat [60].

## 4. Conclusions

This review is limited by the small number of studies evaluating efinaconazole 10% in diverse patient populations, including multiple post hoc analyses of phase 3 trials, which may have resulted in duplicate sampling. Nonetheless, real-world management of onychomycosis is more complex than represented in the clinical trial populations, warranting a tailored, personalized approach (Table 3). In this work, we comprehensively reviewed challenges in treating vulnerable patients and provided practical recommendations. While often perceived as a mild, self-limited infection, onychomycosis is a chronic condition with a risk of antifungal resistance, which is particularly concerning for vulnerable patients with a disproportionate disease burden. Efinaconazole 10% represents an important addition to our therapeutic arsenal, with a favorable efficacy and safety profile across pediatric, elderly, and diabetic populations. This is further supported by its in vitro efficacy against dermatophytes, including terbinafine-resistant strains. Practical considerations when applying topical antifungals include foot examinations to identify tinea pedis, which could disseminate and cause relapse or re-infection of onychomycosis if left untreated. A shared decision-making approach is invaluable for managing onychomycosis. It helps set expectations for treatment response, compliance, and the potential need for long-term treatments (6 to 12 months or more) as well as maintenance therapies. Additionally, patient expectations about visual nail improvements and the use of nail polish should be discussed.

## Figures and Tables

**Figure 1 jof-11-00843-f001:**
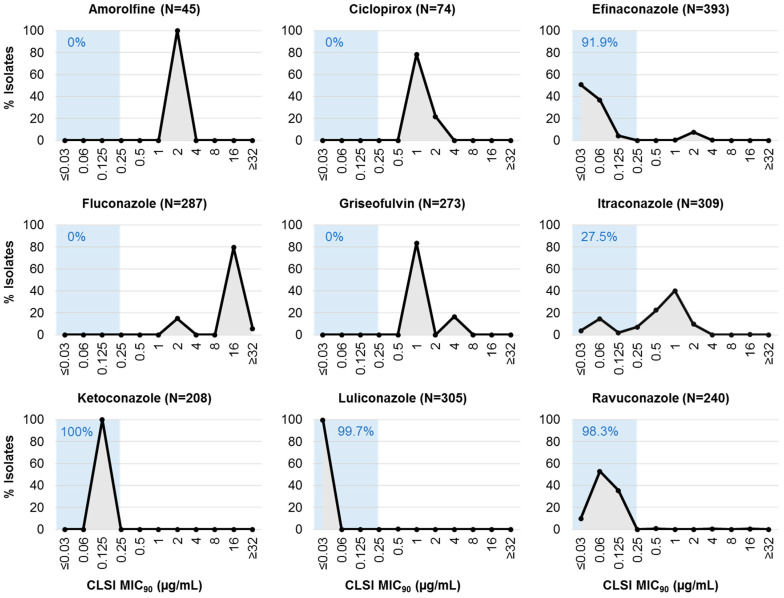
Comparative analysis of in vitro antifungal susceptibility profiles in dermatophyte isolates that exhibit non-susceptibility to terbinafine (MIC_90_: ≥0.25 µg/mL). The blue-shaded box indicates the cumulative percentage of isolates with MIC_90_ values of ≤0.25 µg/mL for amorolfine, ciclopirox, efinaconazole, fluconazole, griseofulvin, itraconazole, ketoconazole, luliconazole, and ravuconazole.

**Table 1 jof-11-00843-t001:** Characteristics of included studies that evaluated efinaconazole 10% in diverse populations.

Study ID	Randomization (Y/N)	Blinding	Diagnosis	Pathogen	No. Patients	Severity
Children (<18 years)						
Eichenfield 2020 [11]	N	N/A	DLSO	Dermatophyte	62	≥20% toenail area affected No matrix (lunula) involvement or dermatophytomas
Elderly (≥65 years)						
Lipner 2025 [12] Gupta 2014 [13]	Y	Double	DLSO	Dermatophyte *	218	36–37% toenail area affected No matrix (lunula) involvement or dermatophytomas
Iozumi 2019 [14]	N	N/A	DLSO SWO	*T. rubrum* TMTISC *Trichophyton* spp.	118	≥20% toenail area affected No proximal nail fold involvement
Diabetic Patients						
Vlahovic 2014 [15]	Y	Double	DLSO	Dermatophyte *	112	36.4% toenail area affected No matrix (lunula) involvement or dermatophytomas
Shofler 2020 [16]	N	N/A	NR	*T. rubrum* TMTISC	40	≥20% toenail area affected
Hispanic/Latino Patients						
Cook-Bolden 2017 [17]	Y	Double	DLSO	Dermatophyte *	270	20–50% toenail area affected No matrix (lunula) involvement or dermatophytomas
Tschen 2013 [18]	Y	Double	DLSO	Dermatophyte *Candida*	135	40% toenail area affected No matrix (lunula) involvement or dermatophytomas
Tinea Pedis Patients						
Markinson 2015 [19]	Y	Double	DLSO	Dermatophyte *	352	20–50% toenail area affected No matrix (lunula) involvement or dermatophytomas

DLSO, distal lateral subungual onychomycosis; N/A, not applicable; NR, not reported; SWO, superficial white onychomycosis; TMTISC, *Trichophyton mentagrophytes*/*T. interdigitale* species complex. * May include mixed dermatophyte and *Candida* onychomycosis.

**Table 2 jof-11-00843-t002:** Comparative analysis of in vitro dermatophyte susceptibility—per MIC_90_ as determined by the standard broth microdilution method (CLSI)—for efinaconazole and other antifungal agents commonly used in the treatment of onychomycosis.

Dermatophytes (%)											
MIC_90_ (µg/mL)	≤0.03	0.06	0.125	0.25	0.5	1	2	4	8	16	≥32
Amorolfine (N = 668)	38.8	0.0	0.0	4.5	4.5	6.6	41.3	4.3	0.0	0.0	0.0
Ciclopirox (N = 717)	0.0	0.0	1.4	42.1	32.8	21.5	2.2	0.0	0.0	0.0	0.0
Efinaconazole (N = 2912)	78.4	12.2	6.7	0.0	1.5	0.1	1.0	0.0	0.0	0.0	0.0
Fluconazole (N = 442)	0.0	0.0	0.0	0.0	0.0	0.0	9.7	4.1	0.0	66.5	19.7
Griseofulvin (N = 577)	0.0	0.0	0.0	0.0	0.0	47.1	45.1	7.8	0.0	0.0	0.0
Itraconazole (N = 1051)	1.5	18.6	19.8	8.6	12.0	35.6	2.9	0.0	0.1	0.2	0.8
Ketoconazole (N = 208)	0.0	0.0	100.0	0.0	0.0	0.0	0.0	0.0	0.0	0.0	0.0
Luliconazole (N = 974)	99.9	0.0	0.0	0.0	0.1	0.0	0.0	0.0	0.0	0.0	0.0
Ravuconazole (N = 264)	14.4	49.6	32.6	0.8	1.5	0.0	0.0	0.8	0.0	0.4	0.0
Tavaborole (N = 512)	0.0	0.0	0.0	0.0	0.0	0.0	39.8	10.5	49.6	0.0	0.0
**Terbinafine Susceptible Dermatophytes (MIC_90_: ≤0.03 µg/mL; %)**											
**MIC_90_ (µg/mL)**	**≤0.03**	**0.06**	**0.125**	**0.25**	**0.5**	**1**	**2**	**4**	**8**	**16**	**≥32**
Amorolfine (N = 270)	95.9	0.0	0.0	0.0	0.0	0.4	3.7	0.0	0.0	0.0	0.0
Ciclopirox (N = 270)	0.0	0.0	3.7	95.9	0.4	0.0	0.0	0.0	0.0	0.0	0.0
Efinaconazole (N = 417)	93.0	2.6	4.3	0.0	0.0	0.0	0.0	0.0	0.0	0.0	0.0
Fluconazole (N = 137)	0.0	0.0	0.0	0.0	0.0	0.0	0.0	0.0	0.0	48.2	51.8
Itraconazole (N = 417)	0.2	31.2	48.2	16.1	2.6	0.0	0.0	0.0	0.0	0.0	1.7
Luliconazole (N = 158)	100.0	0.0	0.0	0.0	0.0	0.0	0.0	0.0	0.0	0.0	0.0
**Terbinafine Non-Susceptible Dermatophytes (MIC_90_: ≥0.25 µg/mL; %)**											
**MIC_90_ (µg/mL)**	**≤0.03**	**0.06**	**0.125**	**0.25**	**0.5**	**1**	**2**	**4**	**8**	**16**	**≥32**
Amorolfine (N = 45)	0.0	0.0	0.0	0.0	0.0	0.0	100.0	0.0	0.0	0.0	0.0
Ciclopirox (N = 74)	0.0	0.0	0.0	0.0	0.0	78.4	21.6	0.0	0.0	0.0	0.0
Efinaconazole (N = 393)	50.6	36.9	4.3	0.0	0.0	0.3	7.6	0.3	0.0	0.0	0.0
Fluconazole (N = 287)	0.0	0.0	0.0	0.0	0.0	0.0	15.0	0.0	0.0	79.4	5.6
Griseofulvin (N = 273)	0.0	0.0	0.0	0.0	0.0	83.5	0.0	16.5	0.0	0.0	0.0
Itraconazole (N = 309)	3.9	14.6	1.9	7.1	22.3	40.1	9.7	0.0	0.0	0.3	0.0
Ketoconazole (N = 208)	0.0	0.0	100.0	0.0	0.0	0.0	0.0	0.0	0.0	0.0	0.0
Luliconazole (N = 305)	99.7	0.0	0.0	0.0	0.3	0.0	0.0	0.0	0.0	0.0	0.0
Ravuconazole (N = 240)	10.0	52.9	35.4	0.0	0.8	0.0	0.0	0.4	0.0	0.4	0.0

Note: Results are presented as composite data and as subgroups for terbinafine susceptible (MIC_90_: ≤0.03 µg/mL) and non-susceptible (MIC_90_: ≥0.25 µg/mL) isolates. Data are shown as percentages of isolates in each MIC category (≤0.03 to ≥32 µg/mL); a color gradient is applied to correlate with the percentage values.

**Table 3 jof-11-00843-t003:** Onychomycosis vulnerable population considerations for treating physicians.

Parameter	Considerations for Special Populations	Recommendations
Diagnosis	Vulnerable populations often are predisposed to nail thickening and discoloration secondary to co-morbidities, trauma and inflammation, which may mimic onychomycosis.	Have a high suspicion for onychomycosis in patients with reduced immunity, frequent trauma, peripheral vascular disease; check frequently for signs of nail/foot infection, as part of the routine evaluation. Complete laboratory fungal diagnosis to rule out fungal infection, if possible; visual evaluation is not sufficient to diagnose nail fungus.
Oral vs. Topical?	Vulnerable populations may have comorbidity risks for oral drug use, and those using polypharmacy may be at higher risk of oral drug interactions.	Oral therapies typically provide better outcomes, but at a higher risk of adverse events, and need for safety bloodwork monitoring. Topical therapies are safer, but have lower rates of cure versus orals. Application requires high patient compliance over long periods, and application may be compromised by those with mobility/flexibility issues.
When is it cured?	Vulnerable populations may be at high risk for reinfection where immunocompromised; nail dystrophy may not resolve, preventing visual confirmation of infection clearance	Obtain laboratory confirmation of infection status from a well-collected nail sample; keep in mind that mycology cultures and microscopy depend upon a good amount of sample from only the infected subungual areas and cultures are frequently false negative even with the best sampling. Consider prolonged therapy, or combination therapy if visual signs and laboratory testing indicate continued presence of infection, if it can be safely used by the patient. Consider prophylactic therapy to maintain improvements/cure as the nail continues to grow out any residual dystrophy.

## Data Availability

No new data were created or analyzed in this study. Data sharing is not applicable to this article.

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
