# Peer review of "Treating Onychomycosis with Efinaconazole: Considerations for Diverse Patient Groups"

_jof, 2025, doi:10.3390/jof11120843_

Round 1

Reviewer 1 Report

Thank you for your efforts and a beautiful summary of this area of clinical care. 

Line 298: I believe you mean 10% and not 1% efinaconazole. 

Author Response

Reviewer 1 Comments

  1. Thank you for your efforts and a beautiful summary of this area of clinical care.

Authors: We would like to thank you for taking the time to review our work.

  1. Line 298: I believe you mean 10% and not 1% efinaconazole.

Authors: Thank you for pointing this out, we have fixed this typing error accordingly.

Reviewer 2 Report

This is an interesting review, addressing the neglected and underestimated clinical diagnosis of onychomycosis.

The review is also interesting from the point of view of the targeted patients, who, due to their distinctive health problems, raise issues of treatment selection and treatment effectiveness.

Nevertheless, for increased clarity and precision, some changes should be performed as follows:

Title:

- Consider replacing “special patient populations” with “selected groups of patients”

- Consider adding “in North America and Japan”

Introduction

-Lines 47-48: “approved in 2014 by the U.S. FDA” – consider specifying that eficonazole 10% is currently available only in USA, Canada and Japan, but not yet in Europe (as it has not been approved by the EMA yet)

Materials and Methods

-Please mention whether the PRISMA guidelines for scoping reviews have been observed or not in this review?

-Lines 70-71: please correct to “EUCAST”, instead of “EUAST”. Why did the authors consult EUCAST for a medication that has not yet been approved by the EMA?

Results and Discussion

-Please describe the limitations of the study; also, emphasize the importance and the novelty of this review.

For increased clarity and precision, some changes should be performed as follows:

Title:

- Consider replacing “special patient populations” with “selected groups of patients”

- Consider adding “in North America and Japan”

Introduction

-Lines 47-48: “approved in 2014 by the U.S. FDA” – consider specifying that eficonazole 10% is currently available only in USA, Canada and Japan, but not yet in Europe (as it has not been approved by the EMA yet)

Materials and Methods

-Please mention whether the PRISMA guidelines for scoping reviews have been observed or not in this review?

-Lines 70-71: please correct to “EUCAST”, instead of “EUAST”. Why did the authors consult EUCAST for a medication that has not yet been approved by the EMA?

Results and Discussion

-Please describe the limitations of the study; also, emphasize the importance and the novelty of this review.

Author Response

Reviewer 2 Comments

  1. Title: Consider replacing “special patient populations” with “selected groups of patients”

Authors: Thank you for your suggestion, we agree that the term “special population” is too broad and may not accurately reflect the content of this manuscript. We have revised the manuscript title as “Treating Onychomycosis with Efinaconazole: Considerations for Diverse Patient Groups”.

  1. Title: Consider adding “in North America and Japan”

Authors: Thank you for your suggestion. Although we agree that specifying the regions where efinaconazole 10% is approved for clinical use is important, we think the title should have more brevity. Additionally, given that efinaconazole 10% is currently under active investigation by the EMA, the content of this article may serve to inform healthcare providers outside of North America and Japan.

Instead, we have instead revised the abstract to specify that efinaconazole 10% is only approved for use in North America and Japan (Lines 14-16).

  1. Introduction: Lines 47-48: “approved in 2014 by the U.S. FDA” – consider specifying that efinaconazole 10% is currently available only in USA, Canada and Japan, but not yet in Europe (as it has not been approved by the EMA yet)

Authors: Thank for your suggestion. We have revised the introduction to reflect the current approval status for efinaconazole 10% (Lines 48-53).

  1. Materials and Methods: Please mention whether the PRISMA guidelines for scoping reviews have been observed or not in this review?

Authors: We have followed the PRISMA recommendations for scoping reviews and have specified this in the Materials and Methods section (Lines 63-65).

  1. Materials and Methods: Lines 70-71: please correct to “EUCAST”, instead of “EUAST”. Why did the authors consult EUCAST for a medication that has not yet been approved by the EMA?

Authors: Thank you for your comment. We have corrected this typing error. The reason for using the search word “EUCAST” was to retrieve results on in vitro susceptibility testing (MIC) (see Section 3.5.), as the European Committee has outlined a standardized protocol for the broth microdilution assay used by many laboratories. The inclusion of this term in our search strategy was specifically to identify studies reporting in vitro testing of dermatophyte resistance against efinaconazole, rather than to examine its clinical use.

  1. Please describe the limitations of the study; also, emphasize the importance and the novelty of this review.

Authors: We agree that study limitations, as well as how this work contributes to the literature, should be highlighted. The conclusions section has been revised accordingly (Lines 359-367).

Reviewer 3 Report

The article is well-written, with the introduction sufficient to describe the current knowledge and highlighting the relevant information. The methodology is also well-described, clearly dividing the information from the studies into clinical and in vitro studies. Nevertheless, I would like to ask if clinical case reports and case series were included. I couldn't find this information in the article, so it's necessary to mention whether they were included, and if not, why.

Regarding the results and discussion, I believe the article contains the necessary information. However, these two sections are not divided as the journal template specifies. This needs to be verified.

The conclusions refer to Table 3, which is not the usual approach, as it should simply summarize the findings.

Regarding the results and discussion, I believe the article contains the necessary information. However, these two sections are not divided as the journal template specifies. This needs to be verified.

Author Response

Reviewer 3 Comments

  1. The article is well-written, with the introduction sufficient to describe the current knowledge and highlighting the relevant information. The methodology is also well-described, clearly dividing the information from the studies into clinical and in vitro studies. Nevertheless, I would like to ask if clinical case reports and case series were included. I couldn't find this information in the article, so it's necessary to mention whether they were included, and if not, why.

Authors: Thank you for taking the time to review our work. We would like to clarify that we did not specifically exclude case reports or case series during our search. All retrieved studies were assessed for data matching to vulnerable patient groups, from which five groups were identified with sufficient number of patients: children, elderly, diabetic patients, Hispanic/Latino patients, and tinea pedis patients. We were unable to include case reports or case series in this review due to lack of consistent reporting of patient characteristics, confirmatory testing, and treatment endpoints.

  1. Regarding the results and discussion, I believe the article contains the necessary information. However, these two sections are not divided as the journal template specifies. This needs to be verified.

Authors: Thank you for your comment. In this manuscript, we chose to combine the Results and Discussion sections to avoid repetitions and maintain brevity. It is our understanding that the manuscript layout can be modified to improve readability, as seen in other articles published in this journal.

  1. The conclusions refer to Table 3, which is not the usual approach, as it should simply summarize the findings.

Authors: Thank you for your comment. In the Conclusions section, we aimed to provide simplified, take-home messages for busy physicians who may not have time to read the main body of the article. It is our hope that this approach would improve the utility of our work.

Round 2

Reviewer 2 Report

The authors responded to my earlier remarks and performed the necessary changes.

The authors responded to my earlier remarks and performed the necessary changes.